# Smart Elderly Care Services in China: Challenges, Progress, and Policy Development

Jason Hung 

Department of Sociology, The University of Cambridge, Cambridge CB3 0SZ, UK; ysh26@cam.ac.uk;
Tel.: +44-(0)-7478119080

**Abstract:** In 2017, the State Council of China published an action plan for the construction of a smart and healthy elderly care industry (2017–2020). The action plan designed and implemented by the State Council of China demonstrates the Central Government's determination to informationalise and digitalise the Chinese society. Therefore, the market of smart home care services should expectedly mushroom in the coming decades, as the demand for smart home care increase. However, there are a range of barriers to achieving the massification of smart home care services, which will be discussed in the following sections. In addition to the shortage of family care and nursing services, elders being physically and psychologically vulnerable also engenders the Central Government to accelerate the provision of smart home care services to the Chinese elderly population. Here, smart home investment and delivery are necessary when building a sustainable elderly care system. The investment in smart home elderly care can lessen the long-term burden on China's healthcare system as more elders would be able to self-manage their everyday life and minor physical and psychological problems. In this article, the author would critically analyses China's implementation of smart home elderly care services, particularly on the benefits and challenges of technological advancement in elderly care and the advantages and problems of relevant policy development. The author also highlights how the informationalisation and digitalisation in elderly care and policy development enhance the convenience of the elderly populations' everyday life when family care is limited or absent. Additionally, the author assesses what the gaps are in existing smart home elderly care technologies and policy development that need to be addressed by Chinese policymakers to further advance the safety and convenience of the elderly cohorts' living.

**Keywords:** sustainable development; ageing; technological development; policy development; China

## 1. Introduction

China has been concerned about the national growth rate of the elderly population aged 60 or above. The annual growth rate of the elderly population was 3.37 percent in the 1990s, a figure that was nearly three times the then overall population growth rate of the country [1]. According to the China Research Centre on Ageing, a sum of 202 million elderly individuals dwelled in China in 2013, with 23 million of them aged 80 or above [2]. Among the elderly population in 2013, 100 million and 37 million suffered from non-communicable diseases and disabilities, respectively [2]. The provision of accessible, affordable long-term care services to the elderly has therefore become an urgent task that the Chinese Government has encountered [3]. The ageing population is projected to reach 240 million in 2030 (accounting for 16 percent of the national population) and 450 million in 2050 (accounting for 33 percent of the national population) [4]. Such forecasts have alarmed Chinese policymakers, as China will be turning from an ageing to an aged population. An increase in life expectancy and the decline in fertility rate have been attributed to the sociodemographic transformation of the country. The rapid growth of the elderly population will significantly burden the Chinese healthcare system as they are at higher risk of suffering from chronic illnesses and functional disabilities [4]. In geriatrics, aged

populations often endure a wide range of diseases, dysfunctions and cognitive impairment, so the corresponding healthcare services needed for them are substantial [5]. The rise in the elderly population's life expectancy has compounded the burden of the healthcare system in China in the long-term [6].

The socioeconomic transformation, the increase in rural-to-urban migration, and the low fertility rates propel significant difficulties for the urban and especially rural Chinese elderly to receive family care. A shortage of nursing staff within the country fails to satisfy the growing demand for services in elderly care [7]. In China, there were 2.7 million senior populations aged 80 or above in 2020 and the figure will reach over 100 million in 2050. Analysing the Chinese Longitudinal Healthy Longevity Survey, 2011–2014 datasets, Zhang et al. forecasted that 1.2 million of those aged 80 or above will be severely disabled in 2025, and the demand for low-skilled and high-skilled nursing staff will hit 5.6 million and 11.5 million, respectively. There is an expandingly vast gap between the demand and supply of nursing staff, where the shortage of such elderly caregivers is alarming and puts the sustainability of the Chinese elderly care system significantly at stake [8]. It is noteworthy to highlight the conventional Chinese model in the domestic setting known as the Confucian values of filial piety—family members are obliged to help and take care of the elders and cross-generational connections should be tightly formed [9]. However, given the lack of family care and the shortage of nursing staff, the concepts of smart home care are introduced to ensure the elders can better manage their everyday life. Here, the provision of smart home care aims to deliver safety, low costs in health and assistance, and independence when the elders are dwelling at home by themselves [7]. Smart home care means "the application of Internet of things, information technology, big data, cloud computing and other technologies to elderly care in order to provide the elderly with smart care or smart home environment, meet their needs for healthy and independent life, and finally improve their physical and mental health and quality of life" [7]. Joining Japan, the United States, and Western European countries, China has been turning from an agricultural society to an informational society to advance modernisation [9]. China mushrooming its development as an informational society prompts the supply of smart home care services. When more individuals, including those who are disadvantaged, are given opportunities to acquire digital literacy, a larger share of the Chinese population shall become familiar with and accept the use of smart home care services.

The construction of smart cities is a relatively new urban development strategy aimed at arranging and delivering sustainable urban settings through the enhancement of digital connectivity. Smart cities are the environments with an effective integration of human, physical and digital systems within the settings to deliver a sustainable, inclusive and prosperous future for local citizens. [10]. In China, given the ageing challenges mentioned above, the design, establishment and delivery of smart homes are seen as an urgent and critical approach to build a sustainable and inclusive future that accommodate the needs of the growing populations of senior citizens. The emergence of smart homes for elderly care in China occurred in 2008, where such an innovation had undergone through four development stages, namely the seed stage (2008–2011), the start-up stage (2012–2014), the development stage (2015–2016), and the popularisation stage (2017–2019) [2]. In 2017, the State Council of China published an action plan for the construction of a smart and healthy elderly care industry (2017–2020). The information technologies featuring the Internet of things, big data, cloud computing, and mobile Internet represent comprehensiveness, efficiency, accuracy, and wisdom, aiming to facilitate the convenience and advancement of individuals' everyday life [11]. The action plan designed and implemented by the State Council of China demonstrates the Central Government's determination to informationalise and digitalise the Chinese society. Therefore, the market of smart home care services should expectedly mushroom in the coming decades, as the demand for smart home care shall raise. However, there are a range of barriers to achieving the massification of smart home care services, which will be discussed in the following sections. In addition to the shortage of family care and nursing services, elders being physically and psychologically

vulnerable also engenders the Central Government to accelerate the provision of smart home care services to the Chinese elderly population [7]. Here, smart home investment and delivery are necessary when building a sustainable elderly care system. The investment in smart home elderly care can lessen the long-term burden on China's healthcare system as more elders would be able to self-manage their everyday life and minor physical and psychological problems. Moreover, along with enhancing societies' sustainable futures and senior populations' lifestyles, the smart home application can improve the flexibility of the power load and reduce the investments in China's power supply by 1.13–1.19 trillion RMB (approximately 158–166 billion USD) per year [12]. In designing, building, and delivering digital innovations, smart cities have the disposition to better respond to the worldwide urbanisation and ageing trends that raise opportunities but also challenges for how policymakers create sustainable, inclusive, and equitable urban environments [10].

China issued the "National New-Type Urbanisation Plan (2014–2020)," emphasising that the country has been prioritising to practise a new form of urbanisation known as people-oriented urbanisation. Under China's recent priority, the expansion of the use of age-friendly smart home devices has been seen as one of the core policy focuses in order to facilitate the country's sustainable urbanisation [13]. In this article, the author critically analyses China's implementation of smart home elderly care services, particularly on the benefits and challenges of technological advancement in elderly care and the advantages and problems of relevant policy development. The author also highlights how the informationalisation and digitalisation in elderly care and policy development enhance the convenience of the elderly populations' everyday life when family care is limited or absent. Additionally, the author assesses what the gaps are in existing smart home elderly care technologies and policy development that need to be addressed by Chinese policymakers to further advance the safety and convenience of the elderly cohorts' living. Relevant policymaking or policy amendment would be pivotal for China to accelerate its urbanisation and develop more sustainable, inclusive, and prosperous smart, habitable environments designated for its ever-growing senior populations.

## 2. Technological Advancement

In contemporary China, most enterprises designing and delivering smart home services in elderly care can restrictively provide elders simple services, including emergency rescue, smart door locks, and basic online health advice. More complex services such as comprehensive assistance, health management, and home security are rarely available in the market due to technological limitations [7,11]. The lack of technological progression in the elderly care sector should be considered a minor concern in China, especially in villages and lower-tier cities, because the elderly population lack digital literacy to use complex, advanced technologies, so simple services should suffice when helping the elderly cohorts to live a more convenient and safer life. Enhancing senior populations' digital literacy can lower the cost of knowledge and effective information acquisition, improve the efficiency of resource endowment distribution, and achieve a better sense of access. Zhang (2022) analysed the 2018 China Elderly Social Tracking Survey and found that, compared to the eastern, more urbanised regions of China, senior populations' digital literacy in non-eastern areas has significant room for improvement [14]. Here, for Internet-connected rural elderly in China, senior populations' basic digital literacy is at moderate to low levels. Even with the presence of family members' digital guidance, the effect of digital literacy on the rural elderly's sense of access remains moderated [14]. These findings reveal that rural elderly are, to a large extent, digitally (semi-)illiterate in China and hint that such a population lacks the motivation or willingness to develop their digital literacy even with the help of their junior family members. To further demonstrate the elderly's reluctance to develop their digital literacy, Wang and Wu (2021) noted that seniors perceive innovative devices as "too complicated to use," where they intend to reject learning how to use them to avoid the experience of technological anxiety. They are demotivated to memorise complicated operating procedures given their declined working memory. The senior populations have

the disposition to use innovative devices infrequently, and the lack of practice further limits their development of fluent device use [15].

Relative to younger generations, elders are inclined to demonstrate a negative view of learning new things and possess a rather poor learning ability. Therefore, a highlighted challenge of technological advancement in elderly care is the elderly population's lack of willingness to accept smart elderly care products. Moreover, smart elderly care goods are costly while the consumption behaviour of the majority of the elderly cohorts is conservative, further compounding the elders' acceptance of using smart home care services [9]. Here, despite Zhang (2022)'s findings, younger family members with higher digital literacy and acceptance of new technologies can serve as the significant others to influence or encourage the elder family members to use smart home elderly care services. Beyond domestic settings, developing the Chinese elderly's digital literacy is beneficial to their household's financial, and societies' social, burdens in the long term. Zhang and Nedospasova (2022) analysed the Chinese Social Survey 2017 dataset and unveiled that younger Chines elderly—meaning those who just met the official retirement age (i.e., 60 years old)—who are digitally literate are more likely to stay active in labour markets and earn financial incomes for their households. Local societies housing more digitally literate individuals who reach the retirement age also benefit from the lower social burdens as the senior populations who are active in the labour markets depend less on social pensions [16]. Therefore, within and beyond domestic settings, building senior populations' digital literacy can enhance China's attainment of multi-faceted sustainable futures, implicating the need for local Chinese governments to introduce more digital learning programmes or information to Chinese citizens, especially those who are of senior status. Pilot smart elderly care communities should also be arranged by Chinese policymakers to "test the water" and to understand feedback given by the elderly beneficiaries who experience the use of smart home elderly care services. Only when smart gadgets are user-friendly, reliable, and safe would elders show a willingness to use such technologies because they do not want to spend an undue amount of time and effort to master the method of using these technologies *per se* [11]. User-friendliness enables the elderly populations with a lack, or an absence, of digital familiarity and literacy to learn how to use these smart services and devices.

For the elderly, cognitive decline is a major risk factor for disability and death and a primary barrier to their ability to use innovative products. Jin et al. (2019) analysed the Chinese Health and Retirement Longitudinal Studies and found that Chinese elderly using digital devices more frequently can be cognitively stimulated and slow their rate of cognitive decline down. As a result, findings show that major relevant actors, including the Central Government and local governments of China, should actively popularise the use of digital devices among all age groups, in order to enhance the cognitive ability of senior adults and those who are becoming senior adults [17]. The improvement in Chinese citizens' cognitive ability can sustainably facilitate more potential innovative device users to possess the ability to explore digital platforms in the long term. Such a circumstance propels more Chinese citizens, including those who are of senior status, to become active users of digital devices, especially when those devices *per se* are user-friendly.

Elderly care institutions in China usually give the elderly beneficiaries a "smart box," which offers multiple technological services, such as smart medical care and home services. For example, those pressing the "smart medical" button would be informed of the contact details of nearby hospitals; and those clicking the "housekeeping service" button would be given the home service telephone number. More advanced smart home elderly care devices would be incorporated with big data analysis technology and artificial intelligence technology, which help reduce any manual tedious operation [11]. The major barrier to the expansion of such technological advancement is not any technological limitations but the digital literacy of the users themselves. Rural elderly populations have low education levels and few opportunities to be introduced to technologies. Therefore, again, smart home care providers have to translate complicated, advanced technological programming into user-friendly technological devices, in order to allow less-educated urban natives and

even rural natives to accept the use of such devices. Otherwise, the market for smart home elderly services will remain highly restricted, and services providers may not be able to sustain their businesses due to the lack of demand.

As noted, China is facing a significant problem of nursing staff shortage. Smart homes have the potential to make up for the challenge of the shortage of caregiving labour by reducing the costs of manpower and time and providing a more accurate, efficient, and high-quality service for elders. Zeng and Chen (2022) found that, aside from digital literacy, financial concerns are a primary factor hindering Chinese individuals' purchase of smart homes. While respondents mostly agreed that smart homes are convenient by easing their housework burdens and making elders' lives more independent, they could only purchase smart homes if financial circumstances allow. In order to popularise the use of smart homes in China, local governments need to subsidise their citizens who are financially in need to buy such properties. Otherwise, even if Chinese citizens, including those who are seniors, are willing to accept the use of innovative devices domestically, they are financially barred from purchasing smart homes. Such restrictions limit the lifestyle and caregiving transitions of the local Chinese communities and hamper the attainment of sustainable futures [18]. Li and Woolrych (2021) studied the experiences of Chinese urban senior populations on living in smart homes in Chongqing. They found that ample respondents expressed financial concerns about the access to smart home devices. While those respondents learned the importance of incorporating technology in support of their everyday ageing challenges, they believed that the adoption of such smart interventions, owing to the expensive costs, were only made available for the affluent Chinese senior cohorts [10]. Zhang et al. (2020) echoed by arguing that the Chinese elderly are very sensitive to the price of intelligent products. They, concurrently, have higher requirements for the convenience of innovative devices [7]. Unless such devices are affordable and user-friendly, otherwise local governments can hardly develop, not even popularise, the market for elderly-care smart homes.

As earlier as in 2014, the Ministry of Civil Affairs already implemented a national-level smart home for elderly care project in seven nursing institutions. Here, sleep monitoring and falling detection devices and self-service physical examination services have been installed and delivered in nursing institutions. Additionally, in 2014, Shanghai issued *Guiding Opinions on Promoting the Pilot Construction of Liveable Communities for the Elderly* and established smart care centres in 40 pilot communities, providing services such as emergency assistance, security monitoring, and everyday elderly care [7]. It is noteworthy that the digital literacy and financial security of urban elites are much higher than less privileged urban natives and rural citizens. Therefore, more pilot communities should be established in first-tier cities first. If positive outcomes (such as smart home elderly care users are willing to learn and are satisfied with the use of, such services) result, then pilot communities can be targeted and arranged in second- and third-tier cities, followed by villages. Piloting should, therefore, be implemented in phases. The progress and outcomes of delivering smart home elderly care services in first-tier cities can serve as a reference for lower-tier cities to take into account during piloting in order for the latter to make any adjustments in the provision of smart home elderly-care services, if needed.

Since different interface standards are used by manufacturers among industries, ample manufacturers of smart products encounter difficulties in data collection. It is necessary for the Central Government to form policies and require manufacturers in different industries to standardise the interface standards used [9]. Setting up pilot communities would allow Chinese policymakers to test which interface standard should be used in a standardised, efficient, and cost-effective manner. Pilot studies should, again, be focused on first-tier cities that house the majority of digitally literate elderly populations, so the operation of piloting could be run efficiently. Globally, voice-user interface (VUI) has gained popularity in recent years. Song et al. (2022) surveyed 420 Chinese senior adults and found that Chinese elders accept the use of VUI owing to the satisfactory levels of perceived usefulness, perceived ease of use, and trust [19]. They suggested that more smart devices in the Chinese market

can apply VUI, a function that allows senior adults suffering from declined physical and cognitive abilities to conveniently interact with innovative products.

To facilitate the development and expansion of smart home elderly care services, alongside ensuring such services can be delivered in a user-friendly fashion, it is important to collect customers' behavioural data and analyse such collected big data by smart devices in order to assist the managers of elderly care institutions to make better elderly caregiving decisions [9]. Big data and deep learning technologies have proven value in China. Here, government authorities and local academic scholars use big data of street view images provided by, for example, Tencent Street View images—an alternative to Google Earth—to monitor Chinese neighbourhoods' safety and social organisations in Beijing [20]. Fang et al. (2020) that argue China should prioritise the use of big data, alongside machine learning techniques, to monitor disease patterns suffered by those who need the elderly and long-term care at the population level. They point out that the use of big data and machine learning techniques is increasingly necessary to transform China into a hub allowing for sufficient pro-elderly caregiving and medical support. These technological advancements help build more sustainable futures for Chinese communities in the next decades [21]. However, to date, most smart products offered by Chinese elderly care institutions fail to satisfy the personalised demands of elders themselves since the data collected are significantly insufficient [9,11]. Therefore, Chinese policymakers should create more pilot studies, allowing more elderly cohorts to experience the use of smart home elderly care services while enabling more elderly care institutions to collect customers' data to better personalise the elderly caregiving services in the long run.

Moreover, elderly care institutions should use wearable devices to collect the physiological data of the elders and to transfer such data to the health services institutions via the smart elderly care services platform. Medical professionals can therefore better understand the physical conditions of the elderly populations by analysing the collected physiological data. The availability of physiological data can also allow nursing staff to adjust their services, diet plans, and rehabilitation treatment for the elderly beneficiaries in order to enhance the user-friendliness and personalisation of medical and caregiving services for the elders [11]. This argument shows how behavioural data are the cornerstone to changing the healthcare provision in relation to elderly care. Local governments, guided by their central counterpart, should urge more smart home elderly care providers to collect behavioural data and to circulate such data within the medical and nursing departments or institutions. Therefore, both elderly care and healthcare services quality can be improved. An example of Chinese elders using physiological characteristics to instruct smart devices domestically is their popular employment of smart television interactions. It is difficult for Chinese senior populations to type *pinyin* on television with remote control. Dou et al. (2018) found that the elderly in China have the disposition to use smart televisions where they exercise voice search when instructing the television to perform. The study demonstrates that the collection of necessary physiological data is important in Chinese contexts, in order to enable the innovation of smart home devices to make Chinese seniors' lives more convenient [22]. To effectively understand the needs of the elderly population for smart elderly care, it is sensible to carry out in-depth interviews and questionnaire surveys with elderly interviewees. Questions asked can include their views on smart gadgets in entertainment and leisure, home cleaning, social interaction, and medical care, alongside their willingness to purchase and use such devices [11]. Established sociologists and sociodemographers can lead the research teams to work on collecting such in-depth data from the elderly populations as they are experienced in and have a deep understanding of surveying. Data collected should be organised, structured, and trimmed by these research teams, followed by being shared with the targeted parties or the public.

To date, smart home elderly care data are scattered among various organisations [11]. Hu et al. conclude the Central Government should focus on two streams: enhancing human resources training quality and implementing a standardised intelligent elderly care system, so as to ensure the provision of quality elderly care services personnel and data

compatibility [11]. It is necessary for the Central Government to play a role to proactively centralise private smart home care services providers' collection of behaviour data and to share such data publicly with all private or potential services providers in order to ensure those working in the industry can optimise their opportunities to succeed in sustaining their businesses in innovative elderly care. Grounded in the analysis of the Chinese Longitudinal Healthy Longevity Survey 2002–2014 data, Zhang et al. (2018) found that the visualisation of physiological, demographic, psychological, economic, behavioural, and social data helps Chinese researchers and policymakers better evaluate health conditions of Chinese elderly and arrange pro-elderly health policies. This, again, demonstrates the importance of collecting sufficient, multi-faceted big data in Chinese contexts, in order for relevant actors to better arrange care support for the elderly [23]. The systematic, centralised publicisation of such data can also encourage more potential private services providers to join in the market, allowing for a higher degree of healthy competition in the market which helps keep the market prices down and the quality of services in a satisfactory fashion.

It is noteworthy that the concept of a social robot has been introduced in China in recent years. Here, social robots mean "robotic technologies that are designed to interact autonomously with people across a variety of different application domains in natural and intuitive ways, by using the same repertoire of social signals used by humans" [7]. Elders worry about the replacement of human caregiving by social robots. In order to minimise the replacement by social robots, French elders preferred small robots that resemble teapots, so their children or grandchildren would not be aware of such social robots. In doing so, children and grandchildren would maintain the responsibilities of family caregiving [24]. Liu et al.'s study shows that elders prefer caregiving from, human attachment to, and communications with their own family members [9]. It may be more sensible to view social robots as complementary pieces that help the elders to live their everyday life in a more convenient way rather than to deem social robots as caregiving being in replacement for family care. Chen et al. (2020) applied randomised control trials to analyse 103 Chinese residents who were clinically diagnosed with dementia with an average age of 87.2 years old. They discovered that the use of respondents being accompanied by a social robot over a timeframe of 32 weeks displayed lower levels of loneliness and agitation. However, they noted that the elderly respondents' acceptance rate of using social robots was dissatisfactorily low [25]. Due to the urgency of the ageing challenges China is facing, social robots have the potential to be used substantially in order to assist the everyday life of elders. It is concerning when elders express their anxiety and fear about social robots, given the circumstance that such an innovative technology may be widely applied in the coming years or even decades. Such a negative view faced by elders may be particularly common among rural Chinese elderly populations who are seemingly more conservative but perhaps less independent. It is, therefore, necessary for local government agencies to deliver community education, including arranging public seminars, to rural elders. Rural elders who are introduced to the benefits of befriending social robots may plausibly be more willing to accept such technologies. In Taiwan, Chen et al. (2020) applied the Chinese version of attitudes towards the use of social robot questionnaire to survey 416 Taiwanese health professionals between November 2017 and May 2018. Survey outputs indicated that elders in caregiving needs are more likely to accept the use of social robots if their attitudes towards such use are more positive. By using social robots elders display better psychosocial well-being [26]. Therefore, the outputs of this Taiwanese study further hint that Chinese government officials should actively change the senior adults' perceptions of social robots, in order to allow the integration of social robots into the local health and care services industry.

Another concerning problem is that the application of social robots is proven to be successful societally in Japan, the United States, and Western Europe [24]. As most research on social robots is deemed a success in advanced countries' contexts, it is uncertain if the social robot quality in China can be as satisfactory as in those advanced western countries due to the former's immature and limited capacity of research and development. Therefore,

more funding should be arranged and delivered by local governments, especially those in rural areas, in order to develop better quality social robots.

In addition, China lacks professional training for rural elderly care services staff. China also faces a shortage of labour services for part-time or professional mutual caregiving. The majority of caregivers are women, so such a circumstance causes inconvenience to meet the caregiving needs of male elders [27]. It is important to recruit more part-time caregivers where middle-aged mothers who have children can spend hours of daytime delivering caregiving when children go to school. Part-time caregiving jobs offer significant flexibility so the recruitment of caregivers should be deemed easier and more appealing. Mobile applications can also be developed that match caregivers and elderly cohorts for caregiving services. It is not a significant issue if rural elders lack digital literacy, because their significant others, such as family members, can act as the persons to coordinate and use the mobile applications to recruit part-time caregivers for them. This kind of bonding is coined "mutual assistance" for the elderly, which helps heighten the availability of social support resources within communities. The mutual assistance model for the elderly is an effective approach to solve the shortage of nursing staff and caregivers in rural areas and, meanwhile, enhance the social connections between members of the communities [27]. The use of Internet services allows individuals to match caregiving beneficiaries with caregivers easily, saving a lot of costs and time from paying agencies to search for suitable candidates who can offer caregiving. Amid the pandemic, with social distancing rules and home confinement rules that may occasionally apply, it is necessary to develop smart home care services where a certain level of social communications and interactions can be maintained. Distance communications also become conveniently available where left-behind elderly populations can use digital devices to easily carry out facetime with their family members. The smart home care providers should continue to better design the smart home care services in order to ensure the elderly cohorts can interact with peers within their communities and healthcare staff so the social bonding they enjoy can be strengthened.

Sun (2013) carried out semi-structured face-to-face interviews with 18 family caregivers of individuals with dementia in Shanghai. He found that role strains and family conflicts are salient stressors discouraging them from undertaking caregiving roles [28]. Findings suggest that the inferior status of caregiving roles often hinders potential job-takers from occupying the available positions. The Central Government and local governments should advertise the importance and required skillsets of caregiving, in order to reduce the general public's negative perceptions of caregiving jobs in the long term. Only by lowering the occupational discrimination against caregivers can China meet the needed supply of caregivers in order to respond to the growing ageing concerns of the country.

Smart devices should provide personalised interaction and services to reduce the loneliness suffered by elders who are left-behind. However, the satisfaction of personalisation of using such devices is yet to be desirable, especially in rural areas [9]. The design of smart devices should be human-centric. For example, when an emergency button is pressed by the smart device user and a caregiving staff is sent to the housing unit of the elder, the elder may be dissatisfied that a stranger comes visit their home. It would be more human-centric if the smart device providers can let the users enter the contact details of several trustworthy caregivers of their choice in the devices, so they will feel more comfortable when someone they know visits their home, when necessary. Moreover, local governments or elderly care institutions can also pay for younger-aged elderly cohorts to take care of older-aged elderly counterparts when the former still have yet to experience physical dysfunction or chronic illnesses that bar them from delivering caregiving. For example, a 65-year-old elder can be paid on an hourly basis to provide caregiving to an 85-year-old counterpart who loses, partial or in full, their physical function.

## 3. Policy Development

In addition to discussing the challenges and progress of innovative technologies in elderly care, it is necessary to discuss the relevant (suggested) policy development so as

to understand the maturity and potential of technological advancement in the industry. Hong et al. found that while 83 percent of middle-aged and elderly Chinese respondents used mobile phones, only 6.5 percent of them had access to the Internet [29]. There is still a significant proportion of elderly populations without Internet access, so such a circumstance hints that there is a large potential for smart home care service providers to engage in the market insofar as local governments can continue to expand the coverage and provision of Internet services. Local governments can consider issuing vouchers for mobile phone and Internet access subscriptions to the elderly populations [29,30]. If local governments are willing to hand out vouchers to elders, poorer Chinese citizens should benefit from the suggested policy the most due to their significant financial limitations. Disadvantaged Chinese elders, in particular, can therefore purchase digital devices that allow their access to digital elderly care and healthcare services.

Zhang et al. argued that elders are sensitive to the price of innovative, intelligent devices. They have high expectations of the convenience of using such intelligent services [2]. Elders may be reluctant to learn about new technologies, especially when they are complicated to use. It is thus needed for local governments, especially rural governments, to marketise the importance and benefits of using elderly care technologies, alongside building elders' digital literacy necessary for the use of such intelligent devices. An interesting study was carried out by Sun et al. (2020) in 13 cities in the Heilongjiang Province of China between May and July 2018. The findings indicated that 38.6 percent of respondents aged at least 60 used the Internet. However, they mostly used the Internet for online dating (74.2 percent), dieting (63.1 percent) and exercising (47.1 percent). Sun et al. called for more endeavours delivered by equipment manufacturers and family members, along with local governments, to encourage Chinese senior adults' use of the Internet for elderly caregiving purposes [31]. Zhang also made a constructive, supporting point by highlighting that more conservative elders cannot accept the use of elderly care smart home services in lieu of caregiving delivered by family members [32]. Therefore, inviting elders to experience the use of smart home care services and carrying out more piloting experiments would be deemed necessary to change the narratives of the use of intelligent devices in more conservative, rural regions. While digital illiteracy maybe recognised as a significant barrier to Chinese senior populations' experience of using smart home devices, Li and Woolrych (2021) found that many urban Chinese elderly respondents reportedly were keen on widening their interests in learning digital knowledge and developing intellectual curiously by exploring the Internet in order to build a deeper sense of connectedness and benefit more from the growth of urbanisation and digitalisation in the long-term [10].

Smart elderly care experience centres should be established in rural communities, where elderly cohorts and their relatives or friends should be invited to experience the use of smart, innovative products [11]. Such a suggested policy is conducive to ensuring more individuals would accept the use of non-traditional services or devices in the long-term, raising the demand for smart home care services that ultimately expands the supply of such services in the market. Until elders experience the convenience of smart home elderly care services, they would be yet to know the comprehensiveness, precision, and alternative advantages of intelligent healthcare and elderly care services *per se* [11]. Therefore, allowing elders to experience intelligent services in person would facilitate their acceptance of new technologies.

China, to date, has built smart elderly care services platforms. However, due to the inadequate demand for such innovative services, ample smart home care platforms, funded by the state, are no longer available [7]. It is needed to raise the demand for smart home services if the Central Government plans to build a thriving, sustainable smart elderly care industry in China. Again, marketing and promoting should be the focal points and baby steps for local governments to expand the smart home care market, along with the demand for such services. China's Ministry of Civil Affairs issued an *Action Plan for the Development of Smart Health and Elderly Care Services* (2017–2020), marking a milestone in specialising national policies on smart home services in elderly care. Specific plans for

service popularisation, technology research, construction of the industry standard, and the development of smart care platforms were employed [7]. While the Central Government has formulated national policies, lower-tier governments need to be able to translate these state guidelines into city-level policies. Facing the problems of bureaucracy, city-level governments may receive multiple orders from upper-level governments, so there could be confusion when lower-tier, city-level governments make their policies [33]. A clearer, more organised governance structure should be arranged to ensure city-level governments understand concretely and concisely what elderly care policies they can exploit. As a status quo, the insufficient top-level design and unified standards force some city-level governments to facilitate smart elderly care by themselves within their own administrative regions, resulting in unsatisfactory compatibility of smart elderly care data and severe duplication of the development of service platforms [11]. China follows a complicated governance structure. It is necessary for upper-tier governments to apply horizontal communications to avoid the delivery of different versions of policy guidelines to the same city-level governments. Otherwise, policy confusion will be significantly caused.

Home-based elderly care services used to be public goods exclusively in China. However, in 2010, the *Construction Plan of Social Elderly Care Service System* (2011–2015) was implemented by the General Office of the State Council where home-based elderly care services become quasi-public goods—sharing the dual characteristics of public and private goods [7]. To date, public long-term care financing is minimal and significantly restricted to supporting welfare recipients and subsidising the delivering of residential care beds and operating costs. Within the public sector, weak quality assurance in the long-term care system is concerning. In response, China has been witnessing an increasing involvement of the private sector in supporting the long-term care system [34]. As elderly care services can now be purchased in the market as private goods, it benefits richer individuals, especially urban natives, who can and are willing to spend financial resources on enjoying better-designed elderly care services. It is necessary for lower-tier governments to offer basic elderly care-related insurance to less advantaged urban natives or rural citizens. Otherwise, the socioeconomic inequalities in elderly care would remain profound. Hu et al. back up the suggestion of further integrating differentiated rural and urban elderly care systems and promoting the pooling of basic elderly care within China [11]. By providing a universal insurance system, Chinese citizens, especially those from poorer or rural origins, can benefit from elderly care services. Otherwise, disadvantaged elders are unable to afford expensive healthcare and elderly costs, particularly those that require innovative technologies to operate, barring the sustainability of the ageing society. Another financial challenge faced by elders in the elderly care sector is the lack of pension entitlement to rural elderly populations [25]. It is essential for local rural governments to provide elderly care services at a subsidised rate or as public goods, in order to ensure those living without pensions can enjoy basic elderly care services at the very least. Specifically, local rural governments should purchase smart elderly care services and distribute them to those in need for free or at a reduced rate. Therefore, the disadvantaged elderly populations can also gain access to innovative technologies in favour of their everyday life. However, rural China is unable to deliver free elderly care services completely with the restrictive fiscal revenues [7]. Therefore, part of the elderly care services has to be pragmatically offered as private goods. Smart elderly care services providers aim at making profits through the supply of innovative services, where such technological advanced services are inclined to be expensive [7]. Hence, only richer urban natives may demonstrate an interest in purchasing smart elderly care services as private goods. If local governments want to massify the exploitation of smart home care services, they should subsidise their citizens earning less than a certain level of wages or incomes to purchase common, simplistic smart home care services; otherwise, the smart home care market would shrink. When richer individuals spend their own financial resources to purchase more technological advanced elderly care services, China can ensure that both higher-end and lower-end smart elderly care markets have a certain degree of demand. Such a circumstance motivates smart elderly

care services providers to participate in the market activities, regardless of whether they focus on delivering higher- or lower-end innovative services.

Public–private partnerships have been formed where local governments and commercial enterprises cooperate to deliver smart elderly care services. Here, governments avoid investing in building and running elderly care institutions, but play a role as a regulator [11,35]. Local governments encourage enterprises to develop and deliver smart elderly care services and assume public responsibilities but do not interfere in the latter's management and operation [11]. This demonstrates how local governments lacking financial resources can still regulate the operation of smart elderly care services within their administrative regions. However, even when private enterprises are given the right to enter the elderly care market to provide relevant services, local rural governments need to ensure their citizens have the financial resources to enjoy smart elderly care experiences. One of the possible solutions in relation to financing is to encourage the Central Government to distribute the financial resources for elderly care to rural and urban governments in a more equal manner, so as to prompt rural elderly populations to enjoy smart elderly care services in a similar fashion to their urban counterparts.

## 4. Conclusions

Despite being an upper-middle-income country, China has been facing entrenched sociospatial inequalities. Ample Chinese provinces, especially those in inner and northwest China, are underdeveloped and lack financing. It is necessary for the Central Government to draw a fairer share of policy focus on suburban and rural regions. Here, financing must be in place in order for city-level governments to develop human resources because low salaries and employment status both bar caregivers and nurses from showing an interest in working in elderly care. When smart home elderly care providers design mobile applications to match caregivers with elders who seek caregiving services, it is essential for the providers themselves to ensure the former is paid at a satisfactory, rewarding rate. Such delivery of caregiving services does not and should not need to be full-time jobs. The caregiving provision should complement caregivers' farming, and caregivers in the market should be compensated financially by the beneficiaries of the services and, more importantly, by local governments which subsidise caregiving per se. The inferior status caregivers are attached to, given the conventional social perception that caregivers are uneducated, unskilled and disdained, should be addressed. Here, more public advertising to present the importance, value and positive images of caregivers, especially when China attempts to build a more age-friendly sustainable future, should be arranged and delivered. Public education at family-, community-, and regional-levels should be encouraged where senior family members, community organisations, and local governments, for example, can give guidance or design freely-accessible workshops for the purpose of enhancing individuals' understanding of and respect for whoever undertaking the roles as caregivers.

Financing needs to be efficiently arranged among the public-private initiatives too. The private sector of elderly care in China is established to some extent. Richer individuals are able to take care of themselves as they can financially invest in obtaining elderly care services as private goods—where private services are inclined to be of better quality and more comprehensive. However, lower-tier, and rural, governments that entail the provision of basic, universal elderly care services to less advantaged Chinese citizens need better financing to offer affordable or free elderly care services. Public financing can be supported by state and public-private initiatives. Many of these initiatives have already been piloted and outcomes can unveil whether vertical and horizontal governance is clearly structured and organised as well as if the quality of innovative technologies available in the elderly care market is satisfactory. Relevant actors in China should take the western or Japanese models as references when designing and performing public-private initiatives, as these pro-elderly models are developed maturely and successfully.

While upper-tier Chinese cities have been undergoing the popularisation stage of building smart homes, villages and towns located in less-developed regions might plausibly

still be experiencing the seed stage, the start-up stage or the development stage. The success of popularising the delivery of smart homes in upper-tier cities can accelerate the rate of innovative elderly caregiving development in lower-tier regions as the latter are able to witness the short- and long-term benefits technological advancement can contribute to the construction of sustainable and organised societies. As a result, insofar as local governments from lower-tier regions are motivated to build age-friendly, pro-elderly services, and when the presence of opportunities for introducing relevant public-private initiatives expands to less-developed locations, the development of smart homes will no longer restrictively concentrate in more urbanised, advanced areas.

For China to realise its policy focus of building a people-oriented urbanisation model, the mentioned policy gaps must be addressed insofar as possible. Challenges encompass opportunities, and vice versa. For example, the author mentioned that Chinese senior populations, especially those who are not city-based, might be digitally illiterate to some extent. However, many of them demonstrate curiosity in, and potential to, learn(ing) digital knowledge and skills. So long as China introduces a wider share of its senior citizens to a variety of smart home devices and makes such innovative products financially accessible, a growing number of its elderly populations will expectedly be showing an increased interest in building their digital repertoire. By then, China can better fulfil its policy focus on building a more sustainable urban and regional development. Not only will cities become more technologically advanced and inclusive but also villages and towns will become more urbanised and habitable too.

**Funding:** This research received no external funding.

**Institutional Review Board Statement:** Not applicable.

**Informed Consent Statement:** Not applicable.

**Data Availability Statement:** Not applicable.

**Conflicts of Interest:** The authors declare no conflict of interest.

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
