# Peer review of "Smart Elderly Care Services in China: Challenges, Progress, and Policy Development"

_sustainability, doi:10.3390/su15010178_

Round 1

Reviewer 1 Report

The subject of the article is very interesting and topical. The policy of care for the elderly is extremely important in today's world. The abstract looks very good. Unfortunately, the article in its current form does not meet the requirements of scientific texts. Notes below: 1. This article is a review. And the remarks about China's policy should be considered accurate. Nevertheless, there is no in-depth study of the literature that would constitute the background for the analysis carried out. It would also be the basis for comparisons. At the moment, it is a free statement rather than a reliable analysis. This is also confirmed by the number of cited publications - 13. While in a theoretical article it should be at least 50-60. 2. As a result, the article lacks constructive discussion. Applications that can complement an existing policy. 3. Apart from criticism, it is necessary to formulate conclusions for the improvement of the described strategies. However, it will not be possible without reference to other publications and solutions in this area.

Author Response

The author has added twenty studies and some 2,000 words throughout the main body to strengthen the arguments made in this review paper. The revised manuscript should present more justified arguments supported by evidence/literature.

  1. This article is a review. And the remarks about China's policy should be considered accurate. Nevertheless, there is no in-depth study of the literature that would constitute the background for the analysis carried out. It would also be the basis for comparisons. At the moment, it is a free statement rather than a reliable analysis. This is also confirmed by the number of cited publications - 13. While in a theoretical article it should be at least 50-60.

In the revised manuscript, the author added shy of 3,000 words with the inclusion of exactly 20 more studies. As the author acknowledges that, in the original manuscript, only four studies were used to make the arguments in the entire main body of the text, so there was a lack of in-depth scholarly discourse presented by the author themselves. In response, in the revised manuscript, the author significantly added arguments and studies’ findings drawn by exisitng scholars to enrich the discussion in the main body of the text, in additon to supporting any arguments made by the auhtor themselves (especially in their conclusion section).

  1. As a result, the article lacks constructive discussion. Applications that can complement an existing policy.

Same as above.

  1. Apart from criticism, it is necessary to formulate conclusions for the improvement of the described strategies. However, it will not be possible without reference to other publications and solutions in this area.

In the revised manuscript, the author added more arguments on the improvement of the described strategies. These arguments were made grounded in the additional scholarly discourse the author engaged in in the main body of the text.

Reviewer 2 Report

I feel that the authors failed to meet the requirements for writing a review article.

Review article - is a type of professional paper writing which demands a high level of in-depth analysis and a well-structured presentation of arguments. It is a critical, constructive evaluation of literature in a particular field through summary, classification, analysis, and comparison.

The review article aims to synthesize the previous literature on a particular topic and to provide a critical analysis.  I do not see a critical author' s analysis in the article.

1.   An abstract should provide the major points or a synthesis of the project;
    subheadings (e.g., purpose, methods, results, and conclusions) should be included, if appropriate;
    the length of the abstract should be 200 words;
    citations should not be included in the abstract;
    acronyms and abbreviations should be included only if used more than once.
2. Introduction
    should discuss basic information about the topic;
    the introduction should address the purpose (research question);
    the text should be written in the present tense.

2. Materials and methods
    should be written in the past tense;
    information needed to repeat the review should be provided;
    search strategies, inclusion and exclusion criteria, data sources and geographic information, characteristics of study subjects, and statistical analysis used should be included.
3.  Results
    The authors should include all results;
    their relevance to the objective should be mentioned;
    results should include heterogeneity of study groups or samples;
    statistical significance should be mentioned.
4. Discussion
    basic information and purpose can be repeated;
    the results and their relevance are clearly and briefly discussed .
5. Conclusions

    This section should discuss the purpose discussed in the introduction. The implications of the findings, interpretations, and identification of unresolved issues should also be discussed.

6. Limitations of the study
    An evaluation of whether the research was adequate to arrive at a conclusion that can be applied to a much larger group, with reasons;
    suggestions for future research should be provided.

7. The article as a whole cannot exist in a vacuum, in isolation from other similar scientific research. It is necessary, as on a map, to show the place of your research in the available knowledge base. The connection to the previous literature is present in the article in at least three places. First, at the beginning, as a brief overview to identify a gap or unexplored niche. Second, in a more extensive review of the literature that goes into the theoretical framework of the study. And the third, at the end of the article, where the results of the study are again linked to the previous literature and their theoretical application is presented.

Author Response

  1. The author has added twenty studies and some 2,000 words throughout the main body to strengthen the arguments made in this review paper. The revised manuscript should present more justified arguments supported by evidence/literature.
  2. In the revised manuscript, sentences were cut to trim the abstract.
  3. Given this is a review rather than an original research paper, the author does not follow the reviewer's guidelines of introduction > materials and methods > results > discussion > conclusions > limitations
  1. An abstract should provide the major points or a synthesis of the project;
    subheadings (e.g., purpose, methods, results, and conclusions) should be included, if appropriate;
        the length of the abstract should be 200 words;
        citations should not be included in the abstract;
        acronyms and abbreviations should be included only if used more than once.

  2. Introduction
    should discuss basic information about the topic;
    the introduction should address the purpose (research question);
        the text should be written in the present tense.

    2. Materials and methods
        should be written in the past tense;
        information needed to repeat the review should be provided;
        search strategies, inclusion and exclusion criteria, data sources and geographic information, characteristics of study subjects, and statistical analysis used should be included.
    3.  Results
        The authors should include all results;
        their relevance to the objective should be mentioned;
        results should include heterogeneity of study groups or samples;
        statistical significance should be mentioned.
    4. Discussion
        basic information and purpose can be repeated;
        the results and their relevance are clearly and briefly discussed .
    5. Conclusions

        This section should discuss the purpose discussed in the introduction. The implications of the findings, interpretations, and identification of unresolved issues should also be discussed.

    6. Limitations of the study
        An evaluation of whether the research was adequate to arrive at a conclusion that can be applied to a much larger group, with reasons;
        suggestions for future research should be provided.

Author reply: In the revised manuscript, the author cut out some sentences in the abstract. The author, however, does not follow the gudeline given by Reviewer Two with respect to mentioning purpose first, methods second, results third and conclusions at last. This is because such a suggested guideline should be designated for orignal research papers but not review papers.

  1. The article as a whole cannot exist in a vacuum, in isolation from other similar scientific research. It is necessary, as on a map, to show the place of your research in the available knowledge base. The connection to the previous literature is present in the article in at least three places. First, at the beginning, as a brief overview to identify a gap or unexplored niche. Second, in a more extensive review of the literature that goes into the theoretical framework of the study. And the third, at the end of the article, where the results of the study are again linked to the previous literature and their theoretical application is presented.

Author reply: In the revised manuscript, the author mentioned major arguments, supported by studies, in the introductory section, the main body, and the conclusion section.

Reviewer 3 Report

It is necessary to correct and supplement the abstract. The abstract is written either in the 1st person plural or in the indefinite article, not written in the 1st person (I would like)

It is necessary to precisely define the reasons and goal of the article in the introduction and in the abstract - why does the author analyze the given issue?

I'm not entirely sure if the type of article is the right one - review. From the content point of view, it is more of a theoretical overview.

Author Response

The author has added twenty studies and some 2,000 words throughout the main body to strengthen the arguments made in this review paper. The revised manuscript should present more justified arguments supported by evidence/literature. Now the revised manuscript should be presented more as a review paper rather than a theoretical overview. Also, in the revised manuscript, the author addresses themselves as a third person rather than a first person.

It is necessary to correct and supplement the abstract. The abstract is written either in the 1st person plural or in the indefinite article, not written in the 1st person (I would like)

It is necessary to precisely define the reasons and goal of the article in the introduction and in the abstract - why does the author analyze the given issue?

I'm not entirely sure if the type of article is the right one - review. From the content point of view, it is more of a theoretical overview.

In the revised manuscript, the author engaged in much richer scholarly discourse on the given contexts, in order to present the paper more as a review than a theoretical overview. Also, the author addressed themselves in third-person rather than first-person. Moreover, the author believes the aim of the review is satisfactorily presented in the introductory section.

Round 2

Reviewer 1 Report

The article was corrected according to my suggestions. It could be published in present form. Congratulations.

Author Response

In my latest manuscript, I addressed all points raised by reviewer 1.

Reviewer 2 Report

My comments and suggestions were ignored by the author. I don't recommend the article for publication.

Author Response

In my latest manuscript, I addressed all points raised by reviewer 2.

Round 3

Reviewer 2 Report

Dear colleagues,

This article was submitted to Special Issue "Smart Technologies for Sustainable Urban and Regional Development". This Special Issue invites papers proposing systematic approaches towards the development of smart digital technologies for the conception, innovation, design, and operation of systems for sustainable urban and regional development. I have written many times and I will write again. The title of the article, the key words, references, and the text of the article itself do not describe the contribution to ... "Sustainable Urban and Regional Development...The conclusion is very weak. It does not reflect useful conclusions and recommendations for the Smart Technologies for Sustainable Urban and Regional Development. My previous critical comments have not been studied , not given a detailed response to the comments with the indication of pages what and how it is corrected.

There are three times in the abstract the phrase "the author would like....". Are these revisions of the comments?!

Author Response

Lines 81-7, lines 108-16, lines 124-6, lines 558-68 were added to address the theme of this Special Issue on "Sustainable Urban and Regional Development." Lines 558-68 were added to consolidate the conclusions in relation to the theme of this Special Issue. To the reviewer's inquiry, "the author would like..." was phrased per another reviewer's request.